# Oxidation induced strain and defects in magnetite crystals

Ke Yuan[1], Sang Soo Lee [1], Wonsuk Cha[2], Andrew Ulvestad [3], Hyunjung Kim [4], Bektur Abdilla[5], Neil C. Sturchio[5] & Paul Fenter [1]

Oxidation of magnetite ($Fe_3O_4$) has broad implications in geochemistry, environmental science and materials science. Spatially resolving strain fields and defect evolution during oxidation of magnetite provides further insight into its reaction mechanisms. Here we show that the morphology and internal strain distributions within individual nano-sized (~400 nm) magnetite crystals can be visualized using Bragg coherent diffractive imaging (BCDI). Oxidative dissolution in acidic solutions leads to increases in the magnitude and heterogeneity of internal strains. This heterogeneous strain likely results from lattice distortion caused by Fe(II) diffusion that leads to the observed domains of increasing compressive and tensile strains. In contrast, strain evolution is less pronounced during magnetite oxidation at elevated temperature in air. These results demonstrate that oxidative dissolution of magnetite can induce a rich array of strain and defect structures, which could be an important factor that contributes to the high reactivity observed on magnetite particles in aqueous environment.

[1] Chemical Sciences and Engineering Division, Argonne National Laboratory, Lemont, IL 60439, USA. [2] Advanced Photon Source, Argonne National Laboratory, Lemont, IL 60439, USA. [3] Materials Science Division, Argonne National Laboratory, Lemont, IL 60439, USA. [4] Department of Physics, Sogang University, Seoul 04107, Korea. [5] Department of Geological Sciences, University of Delaware, Newark, DE 19716, USA. Correspondence and requests for materials should be addressed to K.Y. (email: keyuan@umich.edu) or to P.F. (email: fenter@anl.gov)

Redox cycling of iron (Fe) is an essential chemical process in geo- and bio-spheres[1–3]. Magnetite ($Fe_3O_4$) is an iron oxide mineral with an inverse spinel-type structure containing both Fe(II) and Fe(III)[4]. Magnetite is commonly observed in igneous and metamorphic rocks on Earth and has been found in meteorites and rocks on Mars, and it can also be formed through biomineralization processes[5–7]. In aqueous environments, magnetite plays an important role as a recyclable geochemical battery, where electrons are stored/released in the redox active Fe(II)/Fe(III) couple that serves as the main energy reservoir for Fe-metabolizing bacteria[1,8]. Oxidative dissolution of magnetite releases Fe(II), a critical reductant for immobilizing heavy metals and radionuclides in subsurface environment[9,10]. Both oxidative dissolution and oxidation with increasing $O_2$ fugacity and temperature lead to a decrease in the Fe(II)/Fe(III) ratio from 0.5 in stoichiometric magnetite to 0 in maghemite ($\gamma$-$Fe_2O_3$)[11]. Understanding such a phenomenon can provide insights into controlling the reactivity of magnetite used for groundwater remediation and can benefit the proper interpretation of magnetic fields recorded in its domain structure when the mineral was formed (i.e., a paleomagnetic proxy)[12–14].

Several issues regarding magnetite oxidation mechanisms remain unresolved. Observed oxidation kinetics in both dissolution and heated magnetite crystals are in agreement with a spherical diffusion model involving outward diffusion of Fe(II) to the mineral surfaces[15–17]. This is consistent with a core–shell structure model where a partially oxidized magnetite crystal has a magnetite core and a maghemite-like shell[18]. However, there is limited experimental evidence to test this structure because the crystal structures of magnetite and maghemite are nearly identical: their lattice constants differ by ~1%, and the spinel lattice is preserved, while the extra charge left by Fe (II) vacancies is neutralized by oxidation of Fe(II) to Fe(III) in maghemite[19,20]. Eq. (1) represented the oxidative dissolution of magnetite, where Fe(II) and Fe(III) in different sites are denoted by T (tetrahedral) and O (octahedral) with □ representing the cation vacancies.

$$3(Fe^{III})_T[Fe^{II}, Fe^{III}]_O O_4(Fe_3O_4, \text{magnetite}) + 1/2 O_2 + 2H^+ \rightarrow$$
$$3(Fe^{III})_T[Fe^{III}_{1.66}, \square_{0.33}]_O O_4(\gamma - Fe_2O_3, \text{maghemite}) + Fe^{2+} + H_2O$$
$$(1)$$

Since there is a continuous solid solution between magnetite and maghemite, removal of Fe(II) from magnetite leads to the formation of off-stoichiometric magnetite (Fe(II)/Fe(III) ratio between 0.5 and 0) and lattice contraction[19,20], but three-dimensional (3D) observations of the lattice spacing variation with associated strain and defect structures remain limited[16,17]. Strain and defects in minerals are important factors in controlling kinetics of geochemical reactions, such as mineral dissolution/growth and isotope exchange reactions[21–23]. In particular, off-stoichiometric magnetite can undergo a reverse reaction, i.e., reductive growth in the presence of aqueous Fe(II)[24]. Both reductive growth and oxidative dissolution reactions are controlled by diffusion of Fe(II) in and out of the spinel lattice[25].

Structural characterization of magnetite crystals has been conducted using various experimental methods[8,16,17,26–28]. X-ray diffraction and Mössbauer spectroscopy have been widely used to measure the average unit-cell parameters and stoichiometry of magnetite powders[17,19]. Electron diffraction and surface probe microscopies have resolved ordered cation vacancies and surface reconstructions in magnetite[26,27,29–31]. Spatially resolving the strain structure related by Fe(II) movement could potentially provide new insights into the Fe(II) diffusion pathways and its relationship to diffusion kinetics[32,33]. However, the full 3D strain structures associated with off-stoichiometric domains within individual magnetite crystals cannot be achieved using traditional techniques. Synchrotron-based Bragg coherent diffractive imaging (BCDI) uses a coherent X-ray beam and phase-retrieval algorithms to image the morphology of crystals with high spatial resolution (i.e., tens of nanometers) and internal displacement fields (strain) with subangstrom sensitivity in the displacement magnitude[34–42]. The current BCDI measurement has a time resolution of several minutes, and therefore can be utilized to probe slow dynamic changes in strain and defect distributions within a single nm-scale magnetite crystal, providing an opportunity to gain new insights into its oxidation mechanisms.

Here, we compare the structural changes of magnetite crystals of a few hundred nm in size between two oxidation processes: oxidative dissolution at ambient conditions and thermal oxidation at high temperature (up to 250 °C). We seek to explore how the formation of strain and defect structures during oxidative dissolution of magnetite plays a role in heavy metal–mineral interactions. We find that oxidative dissolution of magnetite in aqueous solutions leads to the formation of nonuniformly distributed strain fields induced by localized lattice contraction/expansion, including the formation/dissociation of dislocation defects. In contrast, thermally oxidized magnetite crystals exhibit strain relief with substantially more homogenous internal strain fields. This stark difference in structural response to oxidation likely results from thermal effects, where high temperatures facilitate Fe(II) diffusion that can heal defects in the spinel lattice.

## Results

**Oxidative dissolution of magnetite in acidic solutions.** Pristine magnetite crystals synthesized under hydrothermal conditions mostly have the morphology of octahedra, as seen by electron microscopy (Supplementary Fig. 1). These crystals were normally less than 500 nm across with face-sharing twins forming chains of interconnected crystals. The synthesized crystals were transferred onto a silicon wafer substrate for X-ray measurements. The measured lattice spacings of the pristine magnetite crystals matched those of stoichiometric magnetite (Supplementary Fig. 2a). Only the crystals that were stable on the substrate over repeated acid treatments are reported here. These crystals were mostly twinned crystals with common faces shared with nearby neighbors, as typical for hydrothermally synthesized magnetite samples[28].

Pristine magnetite crystals were first imaged in air by BCDI and then reacted in a 0.1 M HCl solution for specific time periods. The acid was removed and the same crystal was imaged in air again. This process was repeated to record changes on the crystal as a function of reaction time. Among these, two magnetite crystals having morphologies of a pyramid and octahedron are shown in Figs 1 and 2, respectively. Here, the crystal shape imaged by BCDI corresponds to part of the crystal that has the crystallographic orientation satisfying a Bragg condition (that is, the (311) reflection for this study). This is referred to as the Bragg electron density. For example, a BCDI-determined shape can appear to be a pyramid with a square base, instead of an octahedron, when the upper and lower half of a magnetite octahedron are twinned crystals with different orientations, and therefore one half of the crystal becomes "invisible" to the BCDI measurement.

Both pyramid- and octahedron-shaped magnetite exhibit similar morphological changes by oxidative dissolution, primarily shown as volume reduction and surface roughening (Figs 1a, b and 2a, b). The coherent fringes of the 3D diffraction patterns were weakened (Supplementary Fig. 3), primarily due to increases in disorder (e.g., surface roughening). We also find an apparent loss of magnetite volume close to the crystal surface as indicated

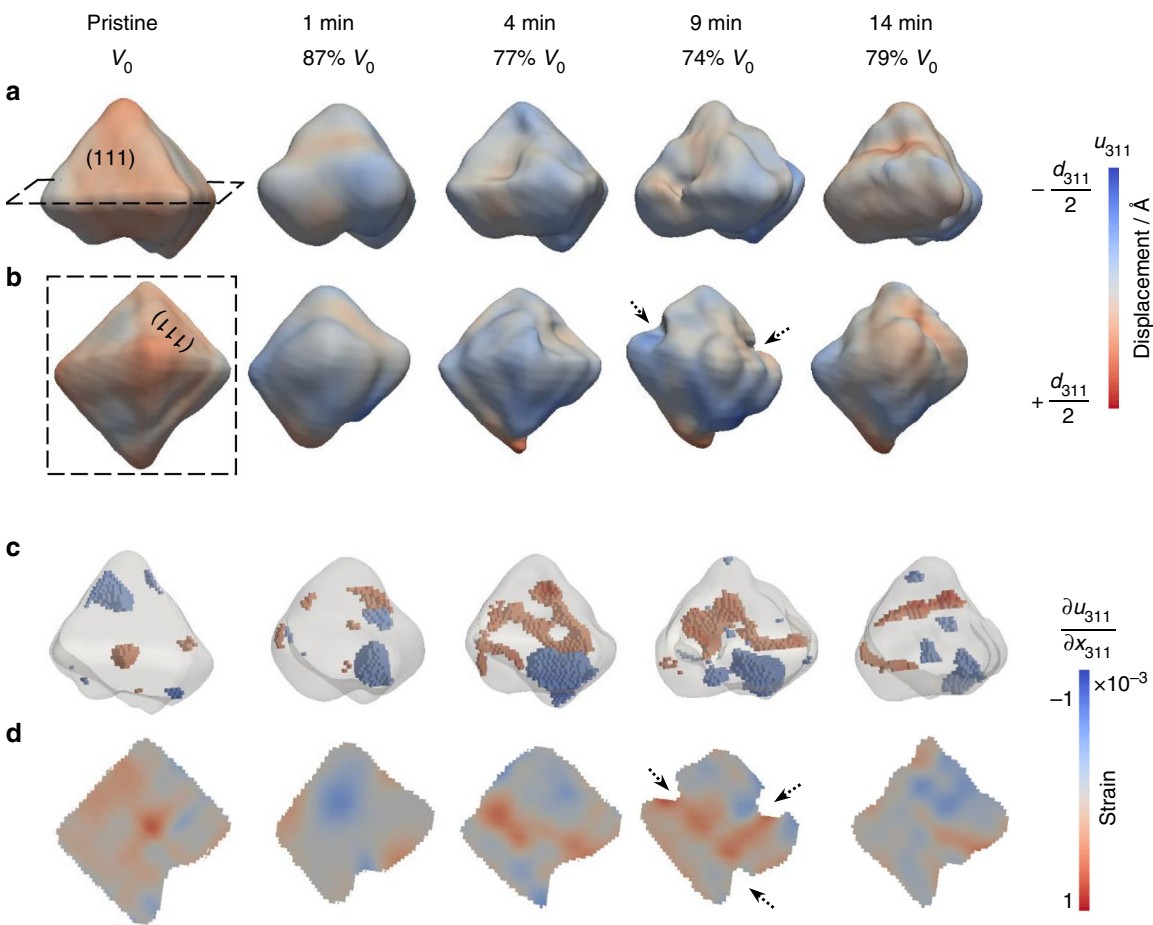

**Fig. 1** Dissolution of a pyramid-shaped magnetite crystal observed ex situ after reaction in 0.1 M HCl solution for 14 min. **a**, **b** were the 3D volume viewed from two different perspectives at a 30% amplitude threshold. The lattice displacements along the [311] direction were projected on the isosurfaces. **c** 3D strain structure associated with compressive (blue, strain < −0.00035 shown in Fig. 3b) and tensile (red, strain > 0.00035 shown in Fig. 3b) strains. **d** Cross-sectional views of the internal strain field of the plane indicated by the dashed lines in (**a**) and (**b**). Magnetite $d_{311} = 2.5314$ Å[43]. Scale bar, 200 nm

by the dashed arrows in Figs 1, 2, and in Supplementary Fig. 4 for another crystal. This volume reduction is likely caused by either dissolution or a localized solid-state transformation from crystalline magnetite to either maghemite or amorphous phases. In contrast, a pair of (111) surfaces indicated by the solid arrows in Fig. 2a, b maintained a relatively smooth surface morphology throughout the reaction (also see Supplementary Fig. 5 from two different views). This is indicated by a pair of relatively long coherent fringes observed in the 3D coherent diffraction pattern at all reaction time steps (Supplementary Fig. 3b). We, therefore, expect that these paired smooth surfaces were likely embedded surfaces shared by nearby twinned crystals (see an example twinned crystal in Supplementary Fig. 6).

In contrast to the similarities in changes of the external morphology, significant differences are observed in the internal displacement fields and strain structures between the pyramid and octahedron-shaped crystals. Increases in strain magnitude and extent were observed after the acid treatment in the pyramid-shaped crystal (Fig. 1c). Here, the strain was calculated from the first order derivative of the displacement field, which is the measured phase (from −π to π) scaled to the units of the lattice spacing (i.e., with values ranging from −$d_{311}$/2 to $d_{311}$/2; here, $d_{311} = 2.5314$ Å for magnetite[43]). Intrinsic strain was observed in the pristine pyramid-shaped magnetite (Fig. 1c, d). Regions of high local strains—both tensile and compressive strains that deviated significantly from the average—increased in the crystal

after 4–9 min reactions, and then decreased slightly in 14 min (Fig. 1c). Regions of high compressive strain (blue), localized mostly at the bottom of the crystal, likely correspond to lattice contraction due to leaching of Fe(II). Removal of Fe was also indicated from the statistical distribution of the local amplitude (as seen in the Bragg electron density histograms, Fig. 3a), where the fractional volume of high amplitudes (i.e., from 0.7 to 1.0) decreased with increasing reaction time. The measured $d_{311}$ spacing of the pyramid-shaped crystal was reduced slightly (by 0.04%) accompanied with increasing FWHM of the (311) Bragg peak (Supplementary Fig. 2), indicative of minor lattice contraction and decreasing crystallinity after dissolution, respectively. Increasing regions of high tensile strain (orange, lattice expansion) were observed in the same crystal (Fig. 1c). However, they were normally separated from regions of compressive strain (blue). These high-tensile strain regions exhibited stripe patterns in 2D (Fig. 1d) and clustered structures in 3D (Fig. 1c).

Changes in strain were more dramatic in the octahedron-shaped magnetite crystal (Fig. 2). Highly strained regions (both compressive and tensile strain) increased more rapidly in magnitude after the same acid treatment compared with those of the pyramid-shaped magnetite (Fig. 2c vs. Fig. 1c and see data outside the dashed lines in Fig. 3b, d). After 1 min of reaction in acidic solution, the crystal exhibited a significant increase in high-strain regions (Fig. 2c, d). The defect structure associated with this high-strain field was extracted using a gradient calculation

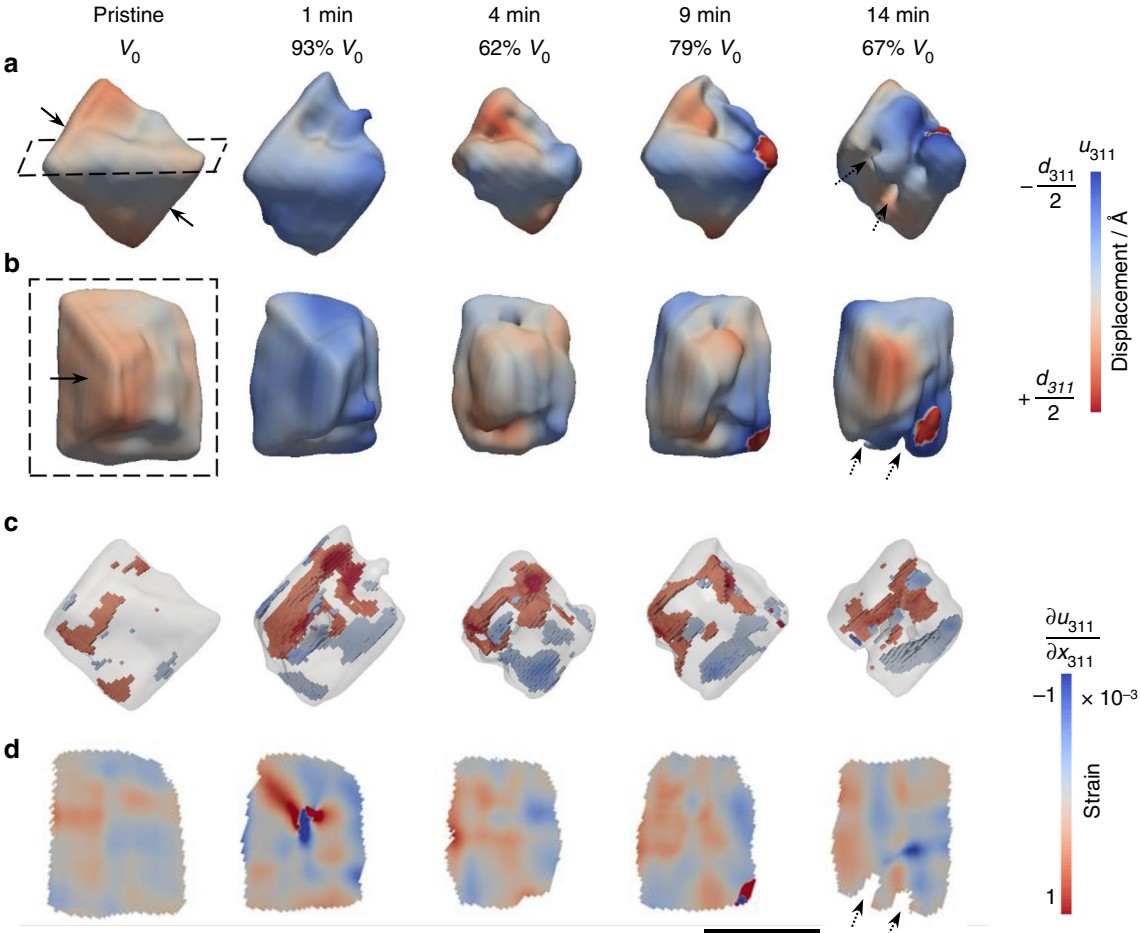

**Fig. 2** Dissolution of an octahedron-shaped magnetite observed ex situ after reaction in 0.1 M HCl solution up to 14 min. **a**, **b** were the 3D volume viewed from two different perspectives at a 30% amplitude threshold. The displacement field along the [311] direction were projected on the isosurfaces. **c** 3D strain associated with compressive (blue, strain < −0.00035 shown in Fig. 3d) and tensile (red, strain > 0.00035 shown in Fig. 3d) strains. **d** Cross-sectional views of the internal strain field at the dashed line box in **a** and **b**. Scale bar, 200 nm

method[36,39]. This analysis reveals a dislocation loop located approximately in the middle of the crystal (Fig. 4a, b). The strain field around this dislocation loop has opposite signs (Fig. 4c), consistent with strain generated from a pair of positive- and negative-edge dislocations associated with a single loop[44]. The coherent diffraction pattern of this magnetite grain (reacted for 1 min) was measured repeatedly (about 18 min measurement time in total) and all measurements showed the presence of a dislocation loop located roughly at the same location (Supplementary Fig. 7), but this loop disappeared with subsequent acid treatments for 3 min (i.e., total 4 min of reaction) (Fig. 2d). The formation of a linear dislocation defect that went through the crystal was observed in another magnetite crystal (Fig. 4d–f) and this linear defect also disappeared in subsequent acid treatments (Supplementary Fig. 8). The crystal shown in Fig. 2a had a 30% decrease in the apparent volume from 1 to 4 min reaction. Considering the Bragg peak intensity of this crystal after 4 min of reaction did not decrease compared with that at 1 min (Supplementary Fig. 2d), and the same amplitude threshold was used for plotting the 3D volume, the volume reduction is likely related to the formation of an amorphous surface layer, which was recrystallized in the subsequent acid treatment (Fig. 2a).

Electron microscopy was performed on the BCDI imaged crystals as well as separately prepared magnetite samples after similar dissolution reactions (Supplementary Fig. 9). The acid-treated crystals after the BCDI experiment maintained their original crystal morphologies under scanning electron microscopy (Supplementary Fig. 9a and b). Some crystals having rounded edges were observed under transmission electron microscopy (Supplementary Fig. 9c and d). The results indicate minor morphological and structural changes after the oxidative dissolution. These observations contrast with the significant changes in the structure and morphology of the BCDI-determined crystals, which is due to the selective sensitivity of BCDI to the crystalline portion of the magnetite satisfying the measured Bragg condition (i.e., reflections from the magnetite {311} planes).

**Oxidation of magnetite in air with increasing temperature.** We now compare the thermal oxidation of three magnetite crystals in air at elevated temperatures (up to 250 °C) with that observed in the low-temperature aqueous dissolution of magnetite. Unlike oxidative dissolution in solution, oxidation in air induced no significant changes on surface roughness and strain for $T < 220$ °C (Fig. 5a–c; also see complete temperature series of this crystal in Supplementary Fig. 10). The volume of the crystal showed no significant change while increasing the temperature up to 250 °C. A partial loss of volume was observed in the crystal shown in Fig. 5a while holding the temperature at 250 °C for 1.5 h. This morphological change is likely related to the phase transformation of magnetite to maghemite, which occurs at ~220 °C[17]. This

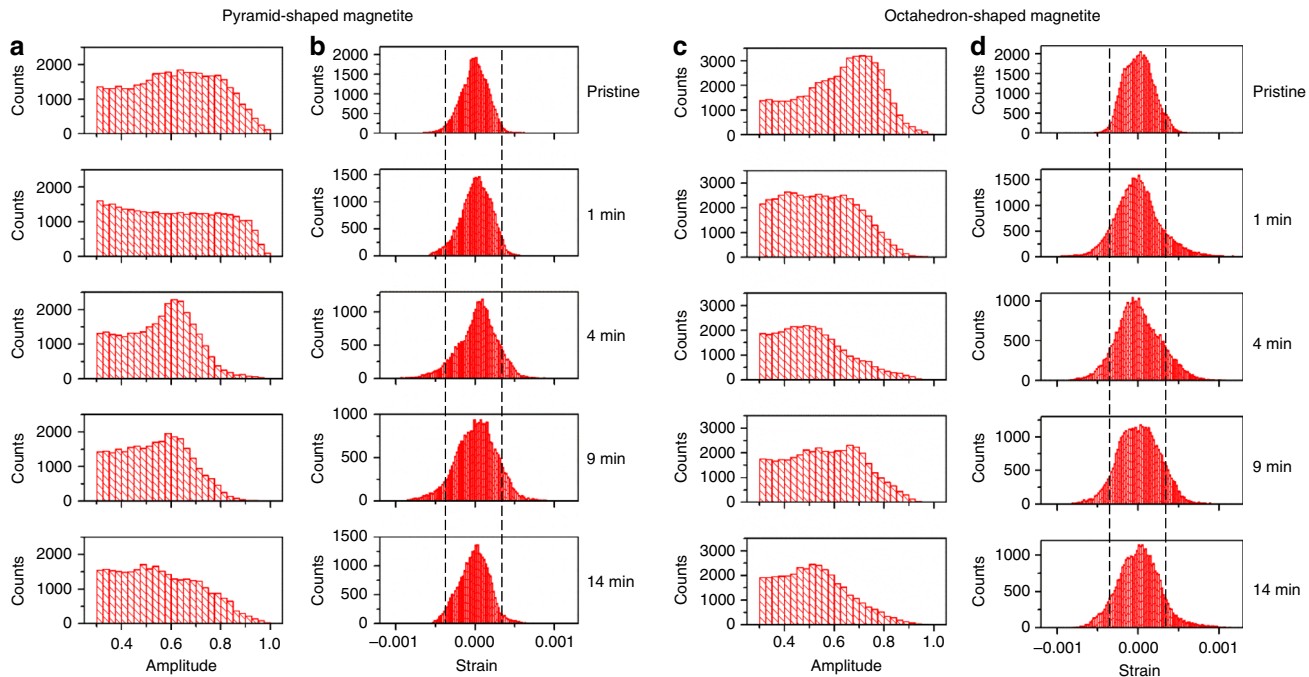

**Fig. 3** Statistical distribution of amplitude and strain of the pyramid- and octahedron-shaped magnetite crystals from Figs. 1 and 2. The amplitude distributions between 0.3 and 1.0 are plotted in **a**, **c** with corresponding strain (**b**, **d**) distributions. Lines on the strain plots indicate the threshold values (−0.00035 and 0.00035) used for plotting the 3D high strain structures shown in Figs 1c and 2c

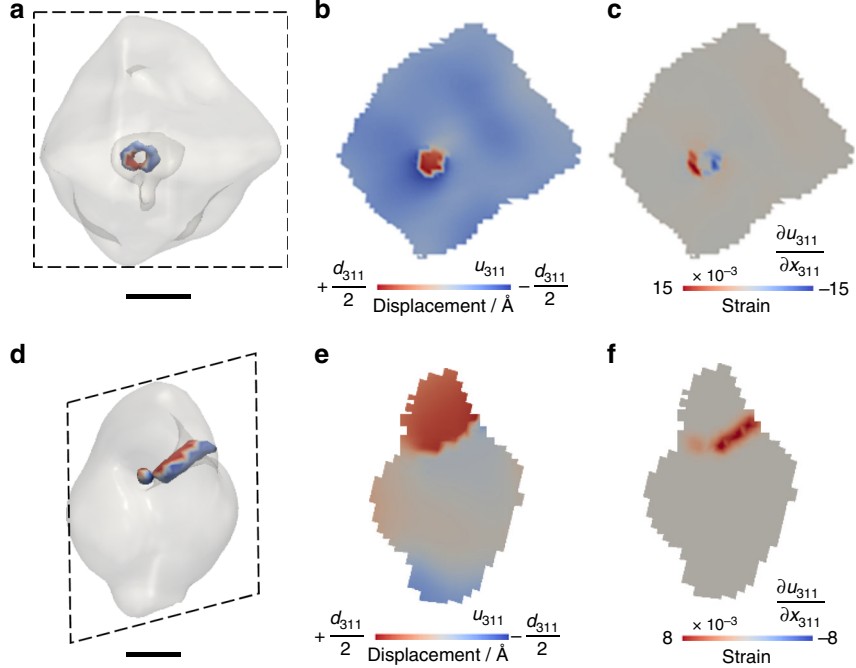

**Fig. 4** Dislocation defects observed during the oxidative dissolution of magnetite. **a** A dislocation loop formed after 1 min reaction in 0.1 M HCl solution of the octahedron-shaped magnetite crystal shown in Fig. 2 with cross-sectional views at the dashed line box showing **b** displacement and **c** strain. **d** A linear dislocation defect formed after 4 min reaction in 0.1 M HCl solution in the magnetite crystal shown in Supplementary Fig. 8 with cross-sectional views at the dashed line box showing **e** displacement and **f** strain. Scale bar, 100 nm

phase transition was also confirmed by changes in the measured $d$ spacing. Magnetite $d_{311}$ exhibited a similar lattice spacing from room temperature to 220 °C, followed by an abrupt decrease in lattice spacing above the phase transition temperature and simultaneous increase in FWHM of the Bragg peak (Supplementary Fig. 11). Similar changes in strain were observed in

another smaller magnetite crystal (Fig. 5d), which exhibited a tensile strained core with a compressive strained shell after the same heat treatment in air (Fig. 5e, f). Heating caused strain relief below the phase transformation temperature (see the complete temperature series of this crystal in Supplementary Fig. 12). Neither significant increases in internal strain nor the formation

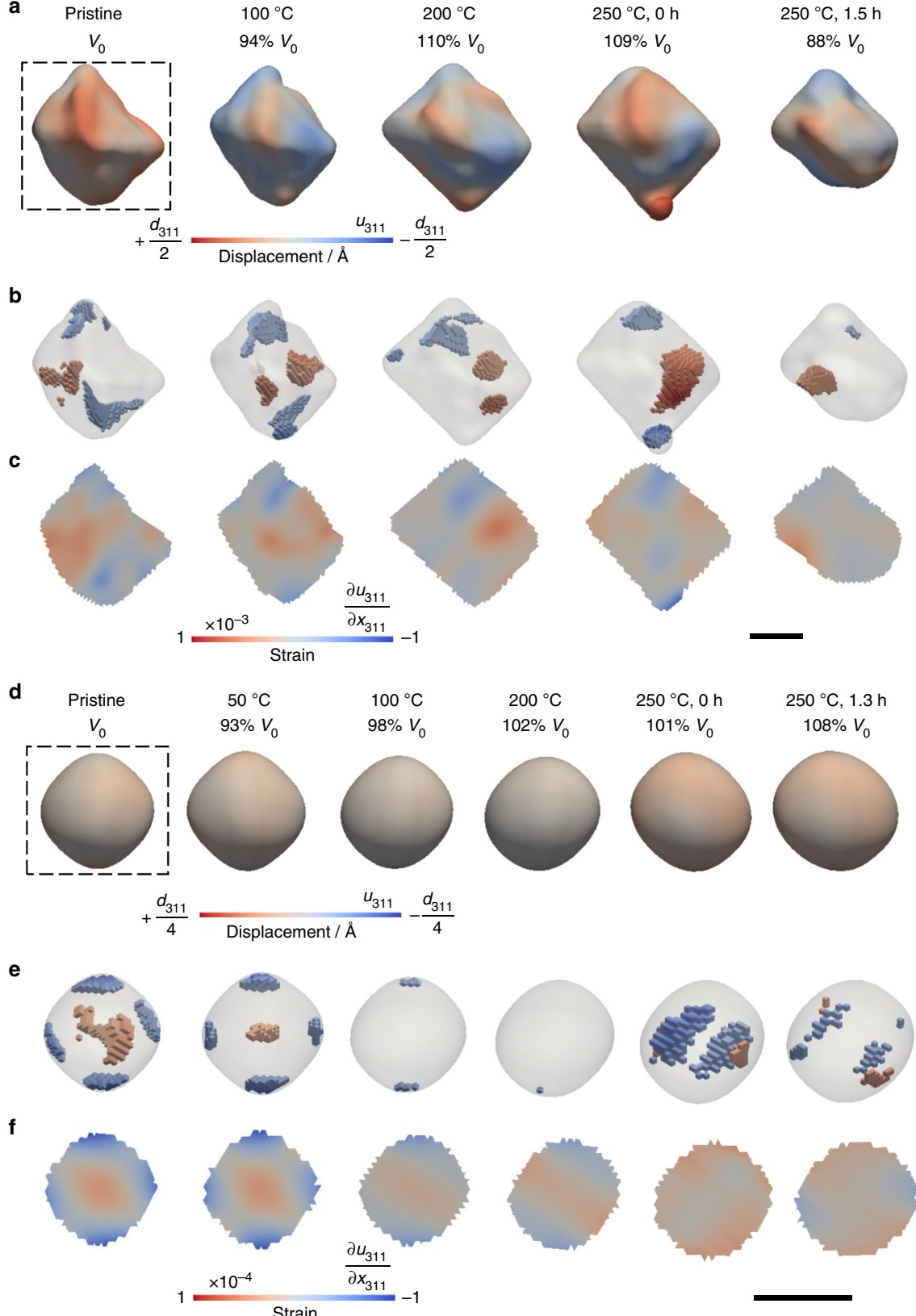

**Fig. 5** In situ heating of two magnetite crystals from room temperature to 250 °C in air. **a**, **d** 3D volume views at a 30% amplitude threshold with lattice displacement along the [311] direction projected on the isosurfaces. **b**, **e** 3D strain structure of compressive (blue, strain < −0.00035/−0.000035) and tensile (red, strain > 0.00035/0.000035) strains. **c**, **f** Cross-sectional views of the internal strain field at the dashed line box in **a** and **d**, respectively. Scale bar, 200 nm

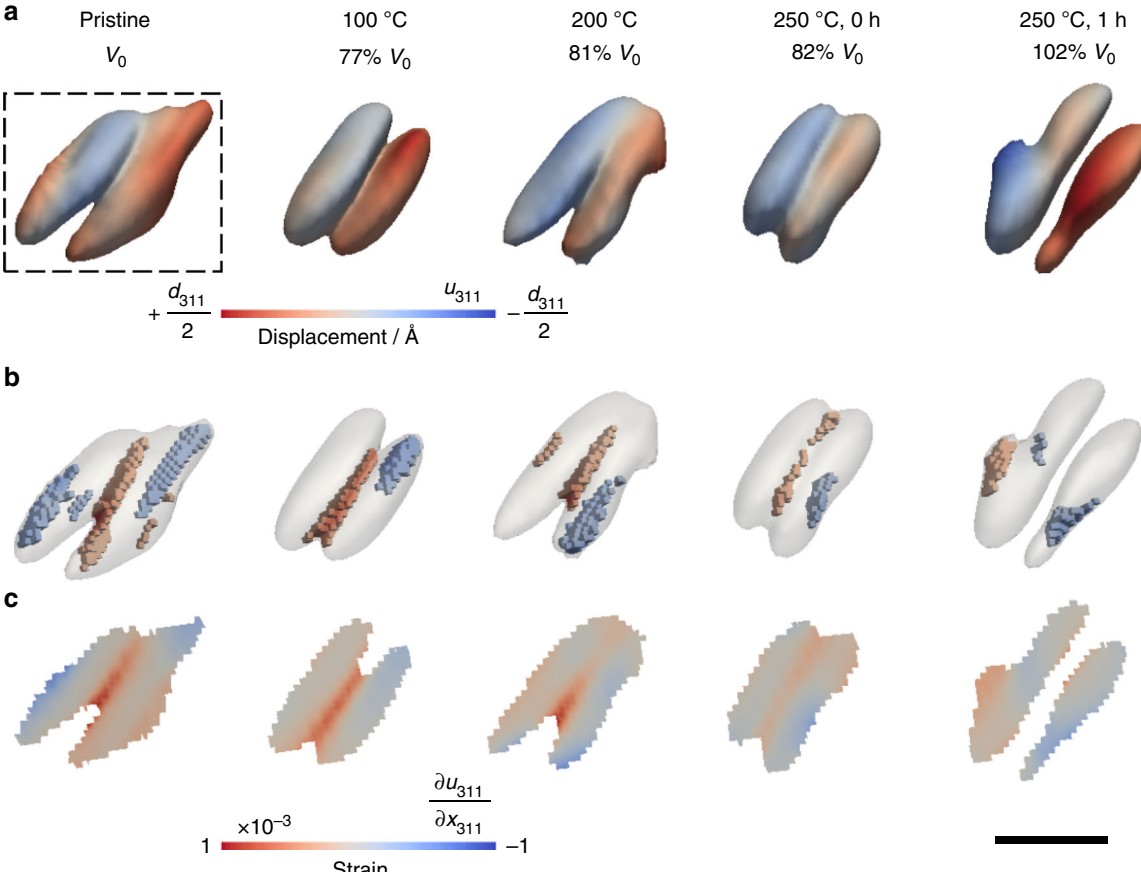

**Fig. 6** In situ heating of a twinned magnetite crystal up to 250 °C in air. **a** 3D volume views at a 30% amplitude threshold with lattice displacement along the [311] direction projected on the isosurfaces. **b** 3D strain structures corresponding to compressive (blue, strain < −0.00035) and tensile (red, strain > 0.00035) strains. **c** Cross-sectional views of the internal strain field at the dashed line box in **a**. Scale bar, 200 nm

of the defect structures were observed during the oxidation of magnetite at elevated temperatures.

Thermal oxidation of a twinned magnetite crystal was also investigated (Fig. 6). The two domains have opposite phase signs (Fig. 6a), presumably resulting from a stacking fault defect at the boundary[28,45,46]. This boundary layer initially had a higher tensile strain than other parts (Fig. 6b, c), and the strain was relieved after heating at elevated temperatures (see complete temperature series of this crystal in Supplementary Fig. 13). Overall, heated magnetite crystals had smaller variations in morphology and strain below the phase transition temperature compared with those observed during the oxidative dissolution in acidic solutions.

## Discussion

Magnetite undergoes a topotactic transformation to off-stoichiometric magnetite and maghemite in oxidation reactions, which involves minimal changes in crystal size, morphology, and structure[15,16,20,47,48]. Here, the measured average $d_{311}$ spacing of individual magnetite crystals in both oxidative dissolution and thermal heating experiments showed minor decreases representing a mild-oxidation process of magnetite. The primary oxidized phases that were present consisted of off-stoichiometric magnetite, i.e., solid-solution phases between magnetite and maghemite. The magnetite crystals reacted in acidic solutions showed not only signs of oxidative dissolution, as indicated by $d$ spacing decrease and volume reduction, but also local reductive growth.

For example, the embedded surfaces showed area decrease/increase (dissolution/growth) during the reaction with acidic solutions (Fig. 2a, b surfaces pointed by arrows, and Supplementary Fig. 5). As mentioned above, these surfaces are likely shared by neighboring twin crystals. Fe(II) liberated during the oxidative dissolution from nearby crystals is likely responsible for the growth. Fe(II) is an active electron donor and can induce recrystallization of iron oxide minerals[2,25,49]. Fe(II) ions likely induced amorphization/recrystallization of part of the existing (111) surfaces, which contributed to the observed surface roughening and volume loss/gain as seen by BCDI. At the same time, Fe(II) adsorbed on magnetite surfaces might diffuse into the spinel lattice which indirectly induced the growth of the embedded surfaces[25].

The measured amplitude (Bragg electron density) and phase (strain) were based on signals from the magnetite {311} planes, which is the most intense Bragg peak of magnetite. Full three-dimensional electron density and strain field maps can be obtained if three independent Bragg peaks from the same crystal are imaged[37]. However, measurements from a single plane are still sensitive to the Fe(II) motion induced structural changes. Magnetite {111} planes are parallel to vacancy sites where Fe(II) atoms can diffuse in the spinel lattice[32,50]. Changes in electron density and strain of the {111} planes can be effectively projected on the measured {311} planes. This helps to explain the observed large range of volume and morphological changes on the amplitude based volume in BCDI compared with that obtained by the electron microscopy. It is likely that other secondary phases,

such as iron (oxy)hydroxides and iron oxides were present during the oxidation of magnetite[17,20,48]. We could only identify these phases if their $d$ spacing values are close to the $d_{311}$ of magnetite and if they were oriented nearly parallel to the $d_{311}$ plane of the imaged magnetite crystal. While these restrictions limited our ability to image other secondary phases, it helped us to focus the measurements on magnetite crystals without the interference from other signals.

The formation of dislocation loop defects has been observed during the dissolution of silver nanoparticles and calcite crystals, phase transformation of palladium crystals, and mechanical indentation of gold nanocrystals by BCDI[40,51–53]. A dislocation loop is a defect structure whose dislocation line is closed inside the crystal[44]. Magnetite, $(Fe^{III})_T[Fe^{II},Fe^{III}]_OO_4$, has oxygen anions forming a face-centered cubic (fcc) lattice with Fe(III) that occupies one-eighth of the tetrahedral sites and Fe(II) and Fe(III) together occupy one-half of the octahedral sites[4]. Defects and dislocations are expected and have been observed in the cubic close packed {111} planes, which are defined by alternating layers of Fe and O atoms[31,54,55]. A dislocation loop can be produced by the collapse of local vacancies and is commonly observed during rapid quenching and displacement cascades generated by irradiation[44,56,57]. As the oxidative dissolution of magnetite involves removal of Fe(II) by creating cation vacancies in the spinel lattice, vacancy accumulation may have occurred and favors a dislocation loop formation within the magnetite {111} planes. A similar observation on the appearance/disappearance of a dislocation loop was also reported during the phase transformation of palladium nanocrystals[52]. Such dislocation loops can be highly mobile as observed in metallic iron under transmission electron microscope (TEM) (diffusion coefficient of around 50 nm$^2$/s)[58,59]. Current BCDI measurements took a few minutes to capture a single coherent diffraction pattern, and therefore do not yet have the appropriate temporal resolution to visualize their movement.

Spatially resolving strain fields and associated defect evolution provide new insights into understanding the oxidation mechanism of magnetite and its role in various geochemical processes. A previous model suggested that partially oxidized magnetite crystals would adopt a simple core–shell structure, where the changes would occur within a thin, but otherwise unobserved, surface layer[17,18]. Our results are ambiguous with respect to whether such a surface shell is formed during oxidation of magnetite as they probe, instead, the evolution of the unreacted magnetite core. The present results highlight the unexpected behavior in which the oxidative dissolution of magnetite induces highly strained regions within the interior of the crystal and can induce discrete dislocations, thereby revealing that such reactions are more structurally complex than previously expected. Numerous studies indicate that strain and defects can influence kinetics in geochemical reactions. For example, Fe(II) can donate electrons to goethite (α-FeOOH) to induce growth/dissolution reactions[2,60]. Such reactions between aqueous Fe(II) and poorly crystalline defect-rich goethite are much faster compared with those occurring on defect-free goethite[21]. In another example, an Fe isotope exchange reaction was observed between aqueous Fe(II) and magnetite[25]. The dissolution induced defects observed by BCDI may facilitate isotope exchange reactions in magnetite, which can influence the reactivity of magnetite used for environmental remediation and the proper interpretations of stable Fe isotope data extracted from magnetite[14,61]. In addition, nano-sized magnetite has been used to sequester heavy metals, such as U, from groundwater[14]. Strain and defects formed during the initial dissolution of magnetite may contribute to the fast metal removal rate observed at the earliest stage of U–magnetite interaction, in addition to factors such as surface area and

stoichiometry of magnetite[9,62,63]. In general, BCDI provides a new opportunity to expand investigations on the influences of strain and defects on mineral reactivity to a large number of aqueous geochemical reactions.

## Methods

**Sample preparations.** Magnetite crystals were synthesized using a wet chemistry method[4]. In brief, 0.3 M Fe(II) solution (FeSO$_4$·H$_2$O, Sigma-Aldrich) prepared in N$_2$ degassed deionized (DI) water was heated to 90 °C in N$_2$ atmosphere. 3.33 M KOH and 0.27 M KNO$_3$ was then added with stirring for 5 h. The black precipitate was cooled and washed with N$_2$ degassed DI water. The crystals were centrifuged and dried in N$_2$ atmosphere. A droplet of magnetite-containing methanol solution was dripped on a Si wafer, where particles formed clusters containing aggregated magnetite crystals as the solvent evaporated. Magnetite nanoparticles were stabilized on the substrate by Van der Waals and magnetic forces.

**Solutions and thermal treatment.** Totally, 0.02–0.05 ml of 0.1 M HCl solution was dripped onto well-isolated magnetite clusters to encapsulate all the crystals inside the solution droplet. The acid was then removed at designated reaction times for BCDI measurements in air and was periodically applied/removed to further react the crystals. For the thermal treatment in air, magnetite crystals on Si wafer were mounted on a heating stage, where the temperature was increased from room temperature up to 250 °C at a step of 50 °C. A total of seven crystals were imaged by BCDI, including four crystals after acid treatment and three crystals by thermal heating.

**BCDI experiment and electron microscopy.** Bragg coherent diffractive imaging was performed at beamline 34-ID-C of the Advanced Photon Source at Argonne National Laboratory. Coherent X-ray beam with energy of 9 keV and flux of $5 \times 10^9$ photons/s was focused to 1.37 μm × 0.7 μm (horizontal × vertical) by the Kirkpatrick–Baez mirrors. The detector was preset to a Bragg position satisfying the diffraction condition from the magnetite (311) plane, and the sample stage was then translated horizontally until an isolated Bragg peak from a single magnetite crystal was observed on the detector. The 3D coherent X-ray diffraction pattern in the vicinity of the magnetite (311) Bragg peak ($2\theta = 31.2°$) was measured through rocking curve scan in the range of ±0.3° in steps of 0.01°. The 2D slices of the 3D diffraction pattern were collected with the exposure time of 1–5 s per image using either a PILATUS (Dectris, pixel size of 172 μm, 487 × 195 pixels) or a Timepix (Amsterdam Scientific Instruments, pixel size of 55 μm, 256 × 256 pixels) detector placed 2.5 or 1.4 m away from the sample, respectively. Total data collection time per sample per coherent pattern was about 2–6 min. The diffraction geometry allowed us to measure a well separated diffraction signal originating from a single magnetite crystal among the aggregates of a large number of crystals. Each measurement was repeated 3–10 times. Phase-retrieval results show consistency for consecutive measurements indicating the minor impact of radiation damage, if any.

An iterative phase-retrieval algorithm is used to reconstruct the real space image of a crystal[64]. The amplitude is proportional to the Bragg electron density (for the crystal planes satisfying the chosen Bragg condition), which yields the morphology of the crystalline material[45]. The real space phase is proportional to the lattice displacement field projected along the measured scattering vector, which provides sensitivity to both strain and defect structures[65]. Details on the phase-retrieval algorithms can be found elsewhere[36,66–68]. In brief, error reduction combined with the hybrid input–output algorithm was used for five random starts of the guided analysis. The support was updated using the Shrink-wrap algorithm[69]. Three generations, each with five individual random started phases, were optimized and the model that showed the lowest error was selected from total of 15 reconstructions[70]. The voxel size was 8.14 × 8.14 × 8.14 and 12.33 × 12.33 × 12.33 nm$^3$ with a conservative estimation on the resolution of 25.67 × 25.67 × 25.67 nm$^3$ based on the phase-retrieval transfer function (Supplementary Fig. 14 and Supplementary Table 1). Electron microscopy images of pristine and reacted magnetite were recorded by Hitachi S4700 scanning electron microscope and Philips CM30 TEM.

## Data availability

The data that support the findings of this study are available from the corresponding author upon reasonable request.

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

## Acknowledgments

This work was supported by U.S. Department of Energy (DOE), Office of Science, Office of Basic Energy Sciences, Chemical Sciences, Geosciences, and Biosciences Division under Contract DE-AC02-06CH11357 to UChicago Argonne, LLC, Operator of Argonne National Laboratory ("Argonne"). Argonne, a U.S. Department of Energy Office of Science laboratory. This research used resources of the Advanced Photon Source, a U.S. DOE Office of Science User Facility, beamline 34-ID-C, operated for the DOE Office of Science by Argonne National Laboratory under Contract No. DE-AC02-06CH11357. Use of the Center for Nanoscale Materials, an Office of Science user facility, was supported by the U.S. Department of Energy, Office of Science, Office of Basic Energy Sciences, under Contract no. DE-AC02-06CH11357. The U.S. Government retains for itself, and others acting on its behalf, a paid-up nonexclusive, irrevocable worldwide license in said article to reproduce, prepare derivative works, distribute copies to the public, and perform publicly and display publicly, by or on behalf of the Government. The Department of Energy will provide public access to these results of federally sponsored research in accordance with the DOE Public Access Plan. H.K. thanks the support by the National Research Foundation of Korea (NRF-2014R1A2A1A10052454 and 2015R1A5A1009962). Finally, we thank Dr. Brian Stephenson (Argonne National Laboratory) for helpful discussions and Kyuseok Yun (Sogang University) for data visualization.

## Author contributions

K.Y., S.S.L., and P.F. designed the study. K.Y., S.S.L., W.C., B.A., and P.F. carried out the experimental measurements. K.Y., W.C., A.U., S.S.L., and P.F. did the phase-retrieval analysis. All authors discussed the results and contributed to the interpretation of data. All authors contributed to editing the manuscript.

## Additional information

**Competing interests:** The authors declare no competing interests.

