## [Peer Review File · Nature Communications]

Reviewers' comments:

Reviewer #2 (Remarks to the Author):

I read the manuscript by Ke Yuan and colleagues entitled "Oxidation induced strain and defects in magnetite crystals, it is certainly valuable work covering a topic which can be of interest for a wider community, even beyond mineralogy and geochemistry, since the oxidation of magnetite crystals has been subject of physical and chemical studies for their numerous applications. The oxidation process of Fe₃O₄ is, in fact, investigated since long and the authors properly cited the relevant studies on this topic.

The authors employed Bragg Coherent Diffractive Imaging to provide (visual) evidence (3D-reconstruction) of the morphology and the internal strain fields induced by two distinct oxidation processes (one chemical and another physical).

BCDI has been already successfully applied to image defect structures in different kinds of materials as properly cited by the authors. As stressed by the authors the use of such state of the art technique can potentially provide useful insights in such topic.

The abstract is clear while the introduction is succinct, likely in the interest of respecting space limits, clear and appropriate.

The methodology and the data quality appear robust, the authors used error bars everywhere obviously possible while provided statistical distributions plots for the amplitude and strain values. Overall the results are clearly and logically presented and the microscopy data (SEM, TEM) provide consistent results supporting the diffraction experiments.

The result of the oxidation processes as visualised by BCDI are strain fields whose highest values are found primarily within the crystal rather than near its surface. This is in apparent contrast with the spherical diffusion model for the oxidation mechanism in magnetite which gives rise to a concentration gradient from a magnetite-like core to a maghemite-like shell. However the modest variation of the d₃₁₁ spacing (Fig. 2 SI) suggests that the effects here discussed correspond to a small change in the stoichiometry of the crystal and thus prevents the possibility of a direct comparison with most of the other studies which investigated the oxidation of magnetite to magnetite over its entire range. In this context, if possible, I would like to know from the authors:

- what are the values of the stoichiometry of the pristine crystals and of the oxidised ones?
- a global characterisation of the structure and stoichiometry, is any secondary phase generated also in tiny fraction? I believe this could also help the reader in easily positioning this study within the existing literature.

- At line 111 and following the authors make the hypothesis that the observed volume loss in one imaged crystal can be attributed to a phase transition from magnetite to maghemite or to amorphisation. This suggests that the results here presented provide only indirect information on phase transitions. How their results on the strain fields can be discussed in view of Gallagher's comment [Nature 217, 1118 (1968), towards the end of the paper] on Egger and Feitknecht's results [Helv. Chim. Acta 45 2042 (1962)]?

Finally I found just few typos like, e.g., at line 253 "acidic solutions showed not only show signs". I recommend the authors to carefully revise the manuscript.

Reviewer #3 (Remarks to the Author):

This paper describes the use of Bragg coherent diffractive imaging (BCDI) to map out strain evolution in magnetite crystals as a function of oxidative dissolution and thermally-induced oxidation. The context of this work, as well as the results, are very exciting, and I was fascinated about the ability to map out the dynamics of strain within individual grains, under the influence of both chemical and thermal effects, with relatively high resolution. This work is very interesting as it does provide detailed information on the differences between oxidative dissolution and thermally-induced oxidation, which provides much insight for topic likes redox-cycling and

environmental remediation. The paper is well written, the figures clearly presented, and the main points are convincingly argued, and I feel it adds significant value to continued investigation of the localised behaviour of magnetite grains, which is an interesting material for study and abundant in nature. Hence, I recommend this paper for publication after some minor revisions:

1. In line 98, it stated that 'magnetite crystals reacted in a 0.1 M HCl solution were imaged in air (ex situ)' – I feel it could be clarified better to say here that the particles were reacted with acid for a certain time, the acid was then removed, and the particles were measured in air. I appreciate the original sentence was quite concise, but I felt it lacked the clarity that is only realised once you read the method section.
2. This study shows very informative localised strain maps from individual grains, which provides much added insight into the methods of magnetite oxidation. However, I feel value is certainly added by putting these localised events into context, where the behaviour of a larger sample set or collection of magnetite grains, are still valuable and feel bulk measurements (like the changes in magnetite d311 spacing and FWHM) do still have a place in the manuscript itself.
3. The supplementary information does certainly support the main manuscript, but I felt like I was reading 2 manuscripts to get the full picture. I feel there could be better integration between the documents and perhaps inclusion of some the images in the article file itself.
4. There are some experimental issues that I feel could be clarified. It is not clear to me how they isolate the (311) reflection from an individual grain. Do they identify a grain and then tilt it until it satisfies the bragg condition or do they detect the (311) reflection, and then by default use that grain, and how do they know that reflection is only from one magnetite grain? I feel all of these questions should be better clarified in the method section of the manuscript.
5. The authors show some nice SEM images of the magnetite grain but, for completeness, they could also include some particle size distributions in the Supplementary information.
6. A trivial correction is that they don't include arrows in Supp Fig. 4, whereas they do for Figs. 1b, 1d and 2d, which are mentioned in the same sentence.

Reviewer #4 (Remarks to the Author):

The present study communicates on visualization of strain field and defects in magnetite crystals by Bragg coherent diffractive imaging and its evolution with temperature and acid dissolution. The resulting reconstructions show sensitivity to strain, and several types of defect are identified within the crystals. Imaging of defects with coherent methods is currently very popular, and is challenging due to high requirements from the beamline stability and reconstruction scheme choice, so this study will be interesting for the community. The manuscript represents an impressive amount of work and includes an interesting materials science discussion on the origin and dynamics of defects. However, it misses some elements that can strengthen the manuscript:

- 1) Why is the 311 Bragg chosen? How do the authors orient crystals to be sensitive to the 311 Bragg peak? Do those crystals have some preferential orientation due to the growth on the substrate (it doesn't seem so from SEM, supplementary Fig.1)? It is clear that we would only be sensitive to those who are in Bragg, but how do the authors know that it is the 311 Bragg? Some more explanation would be useful.
- 2) More experimental details are needed: detectors pixel size, # of pix, coherent flux. Authors communicate on the exposure time; it is also useful to specify time per sample/measurement to know the overhead time. This info is useful for those that consider doing time resolved study to assess time resolution of the technique.
- 3) Rocking curve of the investigated Bragg should be shown to assess if the angular range and

resolution is appropriate.

4) When the acid solution was inserted, did the authors observe any crystal rotation? Or during the measurement itself. Please comment.

5) Also, please comment on radiation damage.

Minor comments and suggestions:

1) Supplementary Fig.1 Indicate where theoretical, where experimental data in the caption/title is.

2) Supplementary Fig.3 Better to show diffraction patterns in jet colormap with colorbar for comparison.

3) When mentioning dynamic changes, specify time resolution applicable (line 73).

4) The interpretation of a volume decrease might not be the only option. Does the measurement include the far away tails of the Bragg? Or is there forward CDI data available for comparison (for at least one sample that is more isolated).

5) As authors mentioned, more than one Bragg measurement is required for full 3D information. Did they try to measure and combine several Bragg measurements?

6) Better to mention voxel size, as it is 3D resolution object (line 348).

Overall, the paper is convincing, original and will be of interest to the materials science and coherent imaging instrumentation communities. I would recommend publication after implementing the relatively minor improvements as described. Data analysis, including statistical analysis seems valid. More details are needed on the BCDI acquisition to ensure reproducibility. The paper is clearly written, conclusions are supported by appropriate citations.

Reviewer #2 (Remarks to the Author):

I read the manuscript by Ke Yuan and colleagues entitled "Oxidation induced strain and defects in magnetite crystals", it is certainly valuable work covering a topic which can be of interest for a wider community, even beyond mineralogy and geochemistry, since the oxidation of magnetite crystals has been subject of physical and chemical studies for their numerous applications.

The oxidation process of Fe₃O₄ is, in fact, investigated since long and the authors properly cited the relevant studies on this topic.

The authors employed Bragg Coherent Diffractive Imaging to provide (visual) evidence (3D-reconstruction) of the morphology and the internal strain fields induced by two distinct oxidation processes (one chemical and another physical).

BCDI has been already successfully applied to image defect structures in different kinds of materials as properly cited by the authors. As stressed by the authors the use of such state of the art technique can potentially provide useful insights in such topic.

The abstract is clear while the introduction is succinct, likely in the interest of respecting space limits, clear and appropriate.

The methodology and the data quality appear robust, the authors used error bars everywhere obviously possible while provided statistical distributions plots for the amplitude and strain values.

Overall the results are clearly and logically presented and the microscopy data (SEM, TEM) provide consistent results supporting the diffraction experiments.

The result of the oxidation processes as visualized by BCDI are strain fields whose highest values are found primarily within the crystal rather than near its surface. This is in apparent contrast with the spherical diffusion model for the oxidation mechanism in magnetite which gives rise to a concentration gradient from a magnetite-like core to a maghemite-like shell. However the modest variation of the d₃₁₁ spacing (Fig. 2 SI) suggests that the effects here discussed correspond to a small change in the stoichiometry of the crystal and thus prevents the possibility of a direct comparison with most of the other studies which investigated the oxidation of magnetite to magnetite over its entire range. In this context, if possible, I would like to know from the authors:

Q1: - what are the values of the stoichiometry of the pristine crystals and of the oxidized ones?

A1: We appreciate the positive feedback provided by the reviewer. We are excited to continue the study on the oxidation of magnetite, which has been a fascinating topic since 1950s.

Regarding the stoichiometry of the pristine and reacted samples, we believe that the best measure of the stoichiometry is the measured d_{311} spacing of magnetite. Based on the review article by Gorski et al. [19] (Am Min, 2010), as magnetite oxidizes, the unit cell becomes smaller due to the smaller size of Fe³⁺ compared with Fe²⁺, and the unit cell parameter decreases linearly with the decreasing Fe²⁺/Fe³⁺ ratio. We think the imaged crystals are close to stoichiometric magnetite because the measured d_{311} spacing values are the same as the reported data (Supplementary, Fig. 2a). It will be very helpful to obtain precise measurement on the

stoichiometry of magnetite by Mössbauer spectroscopy. However, it will be difficult to get the stoichiometry of a single magnetite grain by using Mössbauer spectroscopy. Not all crystals within our sample are stoichiometric based on their measured d spacing. Here, we only reported those close to stoichiometric crystals to initiate the dissolution reactions.

Q2:- a global characterisation of the structure and stoichiometry, is any secondary phase generated also in tiny fraction? I believe this could also help the reader in easily positioning this study within the existing literature.

A2: Our data for samples subject to dissolution and thermal heating is limited to measurements of 7 magnetite crystals. In this work, we did not observe Bragg peaks corresponding to other phases during the oxidation, such as hematite and iron (oxy)hydroxides. However, it is entirely possible that those phases were presented as oxidation products of magnetite, as reported in numerous studies. We added these discussions to line 273 as:

“It is likely that other secondary phases, such as iron (oxy)hydroxides and iron oxides were present during the oxidation of magnetite^{17, 20, 48}. We could only identify these phases if their d spacing values are close to the d_{311} of magnetite and if they were oriented nearly parallel to the d_{311} plane of the imaged magnetite crystal. While these restrictions limited our ability to image other secondary phases, it helped us to focus the measurements on magnetite crystals without the interference from other signals.”

Regarding the mild oxidation instead of investigating the oxidation of magnetite to maghemite over its entire range, with modified Line 249 to:

“Here, the measured average d_{311} spacing of individual magnetite crystals in both oxidative dissolution and thermal heating experiments showed minor decreases representing a mild oxidation process of magnetite.”

Q3:- At line 111 and following the authors make the hypothesis that the observed volume loss in one imaged crystal can be attributed to a phase transition from magnetite to maghemite or to amorphisation. This suggests that the results here presented provide only indirect information on phase transitions. How their results on the strain fields can be discussed in view of Gallagher's comment [Nature 217, 1118 (1968), towards the end of the paper] on Egger and Feitknecht's results [Helv. Chim. Acta 45 2042 (1962)]?

A3: Based on the discussion showed above from the reference cited by the reviewer. We add a discussion in line 292:

“A previous model suggested that partially oxidized magnetite crystals would adopt a simple core-shell structure, where the changes would occur within a thin, but otherwise unobserved, surface layer^{17, 18}. Our results are ambiguous with respect to whether such a surface shell is formed during oxidation of magnetite as they probe, instead, the evolution of the unreacted magnetite core. The present results highlight the unexpected behavior in which the oxidative dissolution of magnetite induces highly strained regions within the interior of the crystal and can

induce discrete dislocations, thereby revealing that such reactions are more structurally complex than previously expected.”

Q4: Finally I found just few typos like, e.g., at line 253 “acidic solutions showed not only show signs”. I recommend the authors to carefully revise the manuscript.

A4: The typos at line 253 has been corrected. The manuscript has been carefully reviewed by all co-authors to eliminate grammatical mistakes.

Reviewer #3 (Remarks to the Author):

This paper describes the use of Bragg coherent diffractive imaging (BCDI) to map out strain evolution in magnetite crystals as a function of oxidative dissolution and thermally-induced oxidation. The context of this work, as well as the results, are very exciting, and I was fascinated about the ability to map out the dynamics of strain within individual grains, under the influence of both chemical and thermal effects, with relatively high resolution. This work is very interesting as it does provide detailed information on the differences between oxidative dissolution and thermally-induced oxidation, which provides much insight for topic likes redox-cycling and environmental remediation. The paper is well written, the figures clearly presented, and the main points are convincingly argued, and I feel it adds significant value to continued investigation of the localized behavior of magnetite grains, which is an interesting material for study and abundant in nature. Hence, I recommend this paper for publication after some minor revisions:

Q1. In line 98, it stated that ‘magnetite crystals reacted in a 0.1 M HCl solution were imaged in air (ex situ)’ – I feel it could be clarified better to say here that the particles were reacted with acid for a certain time, the acid was then removed, and the particles were measured in air. I appreciate the original sentence was quite concise, but I felt it lacked the clarity that is only realized once you read the method section.

A1: We appreciate the positive feedback provided by the reviewer. We agree with the reviewer that the present sentences do not reflect how sample was treated before the measurements. We modified line 98 to:

“Pristine magnetite crystals were first imaged in air by BCDI and then reacted in a 0.1 M HCl solution for a specific time. The acid was removed and the crystal was imaged in air again. This process was repeated to record changes on the crystal as a function of time.”

Q2. This study shows very informative localized strain maps from individual grains, which provides much added insight into the methods of magnetite oxidation. However, I feel value is certainly added by putting these localized events into context, where the behaviour of a larger sample set or collection of magnetite grains, are still valuable and feel bulk measurements (like the changes in magnetite d311 spacing and FWHM) do still have a place in the manuscript itself.

A2: We agree with the reviewer that statistical information provides an averaged picture on the behavior of magnetite during oxidation. To date, the BCDI method has been applied to image individual (or at most a few) particles within the coherent volume of the beam (~ 1 μ m). Recent and anticipated extension of coherent X-ray beams to substantially higher photon energies at the Advanced Photon Source will enable these measurements to access the reflections from a large number of crystals simultaneously, which will enable robust statistical analyses with larger ensembles.

Q3. *The supplementary information does certainly support the main manuscript, but I felt like I was reading 2 manuscripts to get the full picture. I feel there could be better integration between the documents and perhaps inclusion of some the images in the article file itself.*

A3: Unfortunately, due to the limited space, we feel that it is necessary to leave those figures, such as SEM images of FWHM data, in the SI.

Q4. *There are some experimental issues that I feel could be clarified. It is not clear to me how they isolate the (311) reflection from an individual grain. Do they identify a grain and then tilt it until it satisfies the bragg condition or do they detect the (311) reflection, and then by default use that grain, and how do they know that reflection is only from one magnetite grain? I feel all of these questions should be better clarified in the method section of the manuscript.*

A4: We preset the detector to a position satisfying the Bragg condition for the magnetite (311) planes, and then translate the sample stage until a Bragg peak from a random crystal “lights up” on the detector. In some cases, we saw clustered Bragg peaks following the arc of a “powder ring” resulting from multiple crystals having similar orientations, but this situation was unfavorable for phase retrieval due to complex interference between the individual coherent diffraction patterns. Therefore, we only used the reflections from isolated crystals that contained 1 or 2 Bragg peaks. The process of looking for and identifying an ideal crystal for these measurements does consume a significant amount of time. We are working on the chemical synthesis to control the distribution and orientation of magnetite crystals. We have modified the method section line 331:

“The detector was preset to a Bragg position satisfying the diffraction condition from the magnetite (311) plane, and the sample stage was then translated horizontally until an isolated Bragg peak from a single magnetite crystal was observed on the detector.”

Q5. *The authors show some nice SEM images of the magnetite grain but, for completeness, they could also include some particle size distributions in the Supplementary information.*

A5: We now included a histogram showing the particle size distribution based on the SEM images in Supplementary Figure 1(c).

Fig. 1. Size distribution of magnetite particles.

Q6. A trivial correction is that they don't include arrows in Supp Fig. 4, whereas they do for Figs. 1b, 1d and 2d, which are mentioned in the same sentence.

A6: We now included arrows showing the missing volume in Supplementary Figure 4.

Reviewer #4 (Remarks to the Author):

The present study communicates on visualization of strain field and defects in magnetite crystals by Bragg coherent diffractive imaging and its evolution with temperature and acid dissolution. The resulting reconstructions show sensitivity to strain, and several types of defect are identified within the crystals. Imaging of defects with coherent methods is currently very popular, and is challenging due to high requirements from the beamline stability and reconstruction scheme choice, so this study will be interesting for the community. The manuscript represents an impressive amount of work and includes an interesting materials science discussion on the origin and dynamics of defects. However, it misses some elements that can strengthen the manuscript:

Q1) Why is the 311 Bragg chosen? How do the authors orient crystals to be sensitive to the 311 Bragg peak? Do those crystals have some preferential orientation due to the growth on the substrate (it doesn't seem so from SEM, supplementary Fig.1)? It is clear that we would only be sensitive to those who are in Bragg, but how do the authors know that it is the 311 Bragg? Some more explanation would be useful.

A1: We choose the 311 Bragg peak because it is the most intense reflection that yields the best signal to noise ratio for these measurements. Based on Bragg's law, the scattering angle of the Bragg reflection is determined by a combination of the X-ray energy and the lattice spacing of the sample.

In order to find such a reflection, we preset the detector to a position based on the Bragg condition satisfying the magnetite (311) plane, and then translate the sample stage until a Bragg peak from a random crystal "lights up" on the detector. We only used isolated crystals that contains 1 or 2 Bragg peaks to start with. The process of looking for and identifying an ideal crystal for these measurements does consume a significant amount of beamtime. We are working on the chemical synthesis to control the distribution and orientation of magnetite crystals. The method section has been modified based on these suggestions as shown in reply to question #4 from reviewer #3.

Q2) More experimental details are needed: detectors pixel size, # of pix, coherent flux. Authors communicate on the exposure time; it is also useful to specify time per sample/measurement to know the overhead time. This info is useful for those that consider doing time resolved study to assess time resolution of the technique.

A2: We used two different detectors for the data collection: PILATUS detector with a pixel size of $172 \times 172 \mu\text{m}^2$ (487×195 pixels); Timepix detector with a pixel size $55 \times 55 \mu\text{m}^2$ (256×256 pixels). Coherent flux at 34-ID-C was about 5×10^9 photons/s. Because the crystals are randomly orientated, it does take significant amount of time to find samples with good Bragg peak signal (around 1 to 2 hours). Once the Bragg peaks are found, the total data collection time per sample was 2 to 6 min. We modified the method section of the manuscript to incorporate these information mentioned above.

Q3) Rocking curve of the investigated Bragg should be shown to assess if the angular range and resolution is appropriate.

A3: We now included the rocking scan curves of all measured magnetite crystals in supporting information. An example plot is showed here:

Fig. 2. Rocking scan curves of the octahedron-shaped magnetite reacted in 0.1 M HCl as a function of time.

Q4) When the acid solution was inserted, did the authors observe any crystal rotation? Or during the measurement itself. Please comment.

A4: We observed rotation and translation of crystals when we made BCDI measurements in the presence of solution, therefore, the sample was reacted, carefully dried and imaged in air. We started by imaging a large number of pristine crystals (~30 crystals) in air and identify a small fraction of those that were stable under the beam and did not wash away after acid treatment for BCDI measurements. Samples imaged on the thermal heating stage were much more stable compared with those reacted in acid.

Q5) Also, please comment on radiation damage.

A5: The BCDI measurements at one sample were repeated multiple times, and phase retrieval results showed consistency from those consecutive measurements, indicating minor radiation effects. This information is now included in the method section.

Minor comments and suggestions:

Q6) Supplementary Fig.2 Indicate where theoretical, where experimental data in the caption/title is.

A6: We have changed to the caption to indicate the meaning of the dashed lines in Supplementary Fig.2 (a).

Q7) Supplementary Fig.3 Better to show diffraction patterns in jet colormap with colorbar for comparison.

A7: We have tested a few color options and found the current black/white colormap provides the best visualization in terms of the coherent tails in 3D. We found the jet colormap is better used for a 2D cross section view of a 3D coherent pattern.

Q8) When mentioning dynamic changes, specify time resolution applicable (line 73).

A8: We changed the sentence to:

“The current BCDI measurement has the time resolution of several minutes, and therefore can be utilized to probe slow dynamic changes in strain and defect distributions...”

Q9) The interpretation of a volume decrease might not be the only option. Does the measurement include the far away tails of the Bragg? Or is there forward CDI data available for comparison (for at least one sample that is more isolated).

A9: Bragg peaks were centered in the middle of the detector to record the full 3D coherent pattern. Coherent fringes having enough signals far away from the center have all been recorded. The coherent patterns shown in Supplementary, Fig. 3 have been cropped from the original image to highlight the Bragg peak, where the original detector area is much larger. This information has been included in the figure caption. Given our current experimental setup and the distribution of the crystals that are not monodispersed, it is difficult to perform comparable measurements of the crystal shape using the transmitted beam.

Q10) As authors mentioned, more than one Bragg measurement is required for full 3D information. Did they try to measure and combine several Bragg measurements?

A10: We did not measure other Bragg peaks from the same crystal due to the limited beamtime that was available for these measurements. Nevertheless, such measurements will be very useful to obtain a map of all components of the strain field rather than the projection along the (311) direction.

Q11) Better to mention voxel size, as it is 3D resolution object (line 348).

A11: Changed to voxel size.

Overall, the paper is convincing, original and will be of interest to the materials science and coherent imaging instrumentation communities. I would recommend publication after

implementing the relatively minor improvements as described. Data analysis, including statistical analysis seems valid. More details are needed on the BCDI acquisition to ensure reproducibility. The paper is clearly written, conclusions are supported by appropriate citations.

REVIEWERS' COMMENTS:

Reviewer #2 (Remarks to the Author):

I read the rebuttal letter from Ke Yuan and colleagues as well as their revised manuscript which has further improved. I acknowledge the large amount of high quality work required to produce the results presented in the manuscript. I would like to take this opportunity to encourage the authors to proceed, if possible, with the study in order to sample the entire oxidation range from magnetite to maghemite. As I pointed out already in the first review round, the small variation of the stoichiometric index (about 0.05%) suggests that while in this context is technically correct speaking of "Oxidation induced strain and defects" the effects here observed could eventually change significantly with the progress of the reaction and consequently also their impact on the chemical processes mentioned by the authors in the introduction.

As already stated in the first round, overall the manuscript appears robust and provides interesting material for the materials science and diffraction imaging communities; I therefore would recommend its publication.

Reviewer #4 (Remarks to the Author):

As I mentioned in my previous review, I think that this manuscript is interesting for BCDI and coherence in general community and presents original results. The corrections that were implemented after the review process seems reasonable. I recommend for publishing.

Oxidation induced strain and defects in magnetite crystals

Reviewer #2 (Remarks to the Author):

Q1: I read the rebuttal letter from Ke Yuan and colleagues as well as their revised manuscript which has further improved. I acknowledge the large amount of high quality work required to produce the results presented in the manuscript. I would like to take this opportunity to encourage the authors to proceed, if possible, with the study in order to sample the entire oxidation range from magnetite to maghemite. As I pointed out already in the first review round, the small variation of the stoichiometric index (about 0.05%) suggests that while in this context is technically correct speaking of "Oxidation induced strain and defects" the effects here observed could eventually change significantly with the progress of the reaction and consequently also their impact on the chemical processes mentioned by the authors in the introduction.

As already stated in the first round, overall the manuscript appears robust and provides interesting material for the materials science and diffraction imaging communities; I therefore would recommend its publication.

A1: We appreciate the insightful suggestions provided by the reviewer. We agree with the reviewer that a study on the full oxidation process from magnetite to maghemite will provide us a comprehensive understanding on the reaction mechanism. We think that the current approach (i.e., using acidic solutions and heating in air) may not fully convert magnetite into maghemite and so more work will be needed to make those observations, which we hope to be able to do in the next couple of years.

Reviewer #4 (Remarks to the Author):

Q2: As I mentioned in my previous review, I think that this manuscript is interesting for BCDI and coherence in general community and presents original results. The corrections that were implemented after the review process seems reasonable. I recommend for publishing.

A2: We thank the reviewer's comments on our work.